# In Vitro Regeneration Protocol for *Securidaca longepedunculata* Fresen., a Threatened Medicinal Plant within the Region of Lubumbashi (Democratic Republic of the Congo)

Magnifique Chuimika Mulumbati [1,*], Mario Godoy Jara [2,3], Louis Baboy Longanza [1], Jan Bogaert [4], Stefaan Werbrouck [5], Yannick Useni Sikuzani [6] and Michel Mazinga Kwey [1]

1   Plant Breeding and Biotechnology Research Unit, Faculty of Agricultural Sciences, University of Lubumbashi, Lubumbashi P.O. Box 1825, Congo; longanza.baboy@ulb.ac.be (L.B.L.); profmazingakwey@gmail.com (M.M.K.)
2   Haute Ecole Provinciale de Hainaut Condorcet—Laboratory of In Vitro Culture, 7800 Ath, Belgium; godoy1104@gmail.com
3   Center for Agronomy and Agro-Industry of the Province of Hainaut. 11, rue Paul Pastur, 7800 Ath, Belgium
4   Biodiversity and Landscape Unit, Université de Liège-Gembloux Agro-BioTech, 5030 Gembloux, Belgium; j.bogaert@uliege.be
5   Laboratory of Applied In Vitro Plant Biotechnology, Department of Applied Biosciences, Faculty of Bioscience Engineering, University Ghent (UGent), 9000 Gent, Belgium; stefaan.werbrouck@ugent.be
6   Research Unit in Ecology, Ecological Restoration and Landscape, Department of Renewable Natural Resources Management, Faculty of Agricultural Sciences, University of Lubumbashi, Lubumbashi P.O. Box 1825, Congo; sikuzaniu@unilu.ac.cd
*   Correspondence: mulumbatic@unilu.ac.cd

**Abstract:** *Securidaca longepedunculata* Fresen. is an overexploited forest species in the Lubumbashi region (south-eastern DR Congo), as its roots are highly valued in traditional medicine. Conventional propagation of this species is affected by seed dormancy and a high mortality rate during early seedling development. To improve on existing methods, we developed an in vitro seed germination protocol. After observing the germination rates, the effects of different doses (0.5, 1, 1.5, and 2 mg/L) of cytokinins (6-benzylaminopurine, kinetin, and meta-topolin) on *S. longepedunculata* seedling development were compared. Our results showed that soaking for 10 min in NaOCl (10%) followed by 5 min in ethanol (70%) effectively reduced the death rate of seeds while increasing the germination rate to almost 77%. The addition of cytokinins improved plantlet growth: a 12.2× increase in the number of plantlets was obtained with 1.5 mg/L meta-topolin, while only a single stem was obtained from the control. The effects of different auxin types on rhizogenesis did not differ significantly. The best recovery and rooting were noted with microcuttings from the basal parts of *S. longepedunculata* plantlets. Finally, the seedlings produced survived during the acclimatisation phase regardless of the type of substrate used. The established protocol provides a means for large-scale production of *S. longepedunculata* plantlets for the restoration of degraded landscapes and agroforestry.

**Keywords:** *miombo* woodland; medicinal plant; plant growth regulators; culture medium; microcuttings; agroforestry





## 1. Introduction

Forests provide many ecological goods and services to humanity [1]. The importance of the ecosystem services provided by forests is hard to overstate: they contribute to global climate stability, support biodiversity, protect soil and water resources, provide livelihoods for local communities, offer spaces for recreation and meditation, and play an essential role in the ecological resilience of our planet. The conservation and sustainable management of forests is therefore essential to preserving these services and ensuring a sustainable future for generations to come. In developing countries, more than 80% of the population depends

on ecosystem services provided by forests [2]. This includes medicinal plants, which are generally characterised by reduced side effects and lower costs [3–5]. Consequently, more than half of the raw materials for modern medicines worldwide are of plant origin [6].

The excessive collection of plants for medicinal purposes can lead to deforestation. In this context, reforestation seems to be a plausible technique for ecosystem restoration [7,8]. However, when resorting to reforestation, it can be difficult to find reliable sources of seeds from native trees or trees adapted to local conditions in many regions [9–12]. Indeed, native species are less used in reforestation due to their low germination capacity coupled with high seedling mortality [13]. Reforestation can also reduce the genetic diversity of new plantations by using seeds from a limited number of trees and often requires the collection of seeds from natural habitats, which can lead to the disruption of existing ecosystems [14]. An alternative to manual seed collection for reforestation is micropropagation, which enables plants to be multiplied from a small sample of selected plant material [15,16]. Micropropagation enables also the rapid multiplication of plants in the laboratory, which is particularly useful for slow-growing or threatened species [17]. It also enables seedlings to be produced without disturbing wild tree populations, thus contributing to biodiversity conservation [18].

Consequently, micropropagation is a valid and reliable method for the large-scale propagation of plant species [19] and can be initiated from various organs [20–22]. However, its success depends mainly on the plant disinfection process [23], which allows the elimination of pathogens before seeding [24]. In addition, the success of in vitro culture depends on the species, cultivar, tissue type, composition of the culture medium, and age of the cultured organ [25,26]. Moreover, cytokinins and auxins are both known to play an important role in the regulation of plant cell proliferation and differentiation according to their types and concentrations [27,28]. Bogaert et al. [29] showed that MemTR (6-(3-methoxybenzylamino)-purine-9-riboside) performed well in the micropropagation of *Petunia hybrida*, while Bairu et al. [30] reported higher multiplication rates with mT ([6-(3-hydroxybenzylamino)purine]) and MemTR cytokinins in *Aloe polyphylla*.

Covering an area of approximately 2.7 million km$^2$ in Africa, the *miombo* woodland supports the survival of local populations [31–33], including those of the Lubumbashi region in the south-eastern DR Congo. In the rural area of the city of Lubumbashi, people attempt to secure their livelihoods by slash-and-burn agriculture and the exploitation of forest products [34–36]. As a result, the *miombo* woodland cover is decreasing at an annual deforestation rate of about 5% around Lubumbashi city [37]. In the Lubumbashi plain, with a total landscape area of 8877 km$^2$, a 7% loss of *miombo* woodland was observed between 2005 and 2011, compared with a 3.7% regeneration [38]. Among the products collected in this region are medicinal plants, including *Sterculia quinqueloba* [39], *Harungana madagascariensis* [40], and *Psorospermum febrifugum* [41].

In particular, *Securidaca longepedunculata* Fresen., a forest fallow species, is used in the treatment of various diseases, including gastrointestinal ailments, sexually transmitted infections, skin infections, fever, pneumonia, toothache, liver disease, bronchitis, rheumatism, and snakebites [42]. The extract of this plant is rich in anti-inflammatory, antiulcer, antianemia, and antiplasmodial properties [43]. The seeds are used for snakebites, whereas the bark is used for stomach problems and as an arrow poison antidote [44]. Because of all these attributes, the organs of *S. longepedunculata* are sold in all the markets of the city of Lubumbashi (population over 3 million), with a profit margin of almost 200% [45]. This situation is increasing the anthropogenic pressure on *S. longepedunculata* and threatens the survival of its individuals collected by uprooting before flowering [46].

Propagation methods that could facilitate the planting of *S. longepedunculata* in agroforestry systems have been identified [47]. Since its seed dormancy, seedling mortality, and the slower growth of young plants limit its generative reproduction [48], in vitro propagation using local ecotypes is beginning to emerge [47]. Indeed, in Nigeria, regeneration of *S. longepedunculata* through mature zygotic embryo culture was improved by using an appropriate concentration of sugar and plant growth regulators [43]. In Ethiopia, Lijalem

et al. [47] revealed that germination and micropropagation of both coated and de-coated seeds of *S. longepedunculata* under different combinations of phytohormones resulted in different responses according to shoot number and length per explant.

Currently, an in vitro germination and micropropagation protocol adapted to *S. longepedunculata* ecotypes does not exist within the region of Lubumbashi.

Therefore, the objective of the present study was to establish an in vitro regeneration protocol for *S. longepedunculata* by identifying an initiation protocol and the appropriate culture medium for the derived seedlings with different types and doses of cytokinins. We tested the hypothesis that meta-topolin is more suitable than the classical benzyladenine to induce a high proliferation rate.

## 2. Materials and Methods

This study was conducted in the city of Lubumbashi (11°27′–11°47′ S and 27°19′–27°40′ E the capital of Upper Katanga Province in south-eastern DR Congo. The annual rainfall in Lubumbashi is about 1200 mm, and the temperature varies between 17 and 33 °C [49]. During the second half of the last century, the average annual temperature was about 20.1 °C [50], but steady warming has been documented [50]. The characteristic soil of the region is Ferralsol, with acidic pH, low mineral nutrient content, and high concentrations of Fe and Al [51,52]. The dominant forest is the *miombo* woodland, which is undergoing severe degradation due to anthropogenic activities [53]. Agriculture, livestock, services, mining, and trade are the main activities in the city [54], where more than 3 million people live [55].

### 2.1. Initiation of S. longipedunculata Seeds

The *S. longepedunculata* seeds used as explants were obtained from the Luishwishi Reserve, a protected concession located in the north of Lubumbashi city (11°29′26.9″ S and 27°35′58.9″ E). Five healthy plants with no evidence of pathogen attack or root harvesting were selected as mother plants for the production of physiologically mature seeds [56]. To reduce anthropogenic disturbances during fruit filling and ripening, the seed plants were regularly monitored [47]. The trial was conducted from July to October 2022, during the dry season in the region. After collection, seed-bearing fruits were transported to the laboratory and placed in a freezer at 20 °C; only fruits of homogeneous size [57] were kept. After sorting, the fruits obtained were soaked in tap water for 48 h and then shelled to eliminate non-viable seeds [58,59]. The selected viable seeds were disinfected in six solutions consisting of sodium hypochlorite (NaOCl 10%) and ethanol (70%) under different soaking times (Table 1), in a completely randomised design with twenty-four replications, from March to April 2022. Thereafter, seeds were rinsed three times for three minutes in sterile distilled water under a laminar flow hood to remove the disinfectant solution [60,61] and transferred to a jar filled with 50 mL of a solid Murashige and Skoog (MS) medium [62] prepared according to the formulation of Mazinga et al. [63]. Seeds of *S. longepedunculata* as explants were sterilised under different soaking times in the disinfection solution. The jars were placed in the culture chamber at a temperature of 25 °C with an alternating light/dark regime of 16 h/8 h [64] and were observed daily for seed contamination and rot and to calculate germination rates [65,66].

**Table 1.** Disinfection solutions *of S. longepedunculata* seeds according to soaking times, decreasing for NaOCl (10%) and increasing for $C_2H_5OH$ (70%).

| Treatment | Sodium Hypochlorite (10%) | Alcohol (70%) | Reference |
|---|---|---|---|
| Treatment 1 | 20 min | 30 S | [67] |
| Treatment 2 | 15 min | 1 min | [56] |
| Treatment 3 | 15 min | 2 min | [68] |
| Treatment 4 | 10 min | 5 min | [27] |
| Treatment 5 | 10 min | 10 min | [69] |
| Treatment 6 | 5 min | 10 min | [70] |

## 2.2. Influence of the Addition of Cytokinins to the Culture Medium

Three types of cytokinins, 6-Benzilaminopurine, kinetin, and meta-topolin, were obtained from the in vitro culture laboratory of the University of Ghent and the Haute Ecole Provinciale de Hainaut in Belgium and applied in four doses (0.5, 1, 1.5, and 2 mg/L). A treatment without cytokinins was used as a control. The different treatments were replicated six times, and the experiment was carried out in a completely randomised design. Indole-3-acetic acid (IAA) was added as a supplement to the auxin/cytokinin hormone balance at a dose of 0.1 mg/L [71], with the exception of the control (no hormone). The culture medium used was that of Murashige and Skoog [61], according to the formulation of Ilczuk et al. [72].

*S. longepedunculata* seeds disinfected after soaking for 48 h in tap water were then shelled and rinsed three times in sterile distilled water under a laminar flow hood [73]. Thereafter, they were inoculated into a 50 mL test tube containing 10 mL of culture medium. Baskets containing 24 test tubes each (with one basket considered as the experimental unit for each treatment) were placed on a shelf in the culture chamber at 25 °C with a 16 h/8 h alternation of light/darkness to allow for better seed germination [66]. The observations included the germination rate and the number of plantlets.

## 2.3. Influence of Microcutting Part and Auxin Type on In Vitro Rhizogenesis of S. longepedunculata

The *S. longepedunculata* microcuttings used in this phase were derived from in vitro proliferation of seeds in a cytokinin-enriched medium (see Section 2.2). Having produced an average of ten plantlets per seed sown, the plantlets were segmented to obtain more rooted vitroplants in a shorter space of time. Microcuttings 1 cm in size [27] with at least one germinal bud were selected and categorised into apical, median, and basal cuttings. These explants were seeded in tubes packed with 10 mL of culture medium enriched with three types of auxins (Indole-3-acetic acid, 1-naphthalene acetic acid, and indole-3-butyric acid) + 0.1 mg/L BAP. Nine treatments were obtained from the combinations of the three microcutting parts in culture media enriched with the three different auxin types. Auxin type was considered as the main factor, while the microcutting part was subordinate to it. The baskets of tubes (six baskets per treatment) were placed in the culture chamber at a temperature of 25 °C, with a 16 h/8 h alternation of light and darkness [64]. In this trial, observations included the rate of recovery and rooting of microcuttings, as well as the number and length of roots [74].

## 2.4. Acclimatisation of S. longepedunculata Vitroplants

The rooted microcuttings were previously washed with distilled water to remove the culture medium. Next, the vitroplants were acclimatised for one month and transferred to a polyethene bag filled with one of three types of substrates (agrarian soil, undisturbed soil, and a mixture of agrarian and undisturbed soil) in a completely randomised design with 12 replicates. Agrarian and undisturbed soils were collected to a depth of 0–50 cm with a hoe at the University of Lubumbashi agricultural farm (11°34′55.66″ S and 27°24′52.97″ E) and Luswishi Reserve, respectively. The seedlings were sown after watering to the field capacity of the substrate, which had a pH of 5.5. Daily watering with 20 cl of tap water was carried out during the acclimatisation phase. Recovery rate (30 days after plantlet planting) and seedling height (60 days after plantlet planting) were the main observations made.

## 2.5. Data Analysis

The data set collected in this study was subjected to a Shapiro normality test, considered to be the most reliable test determining the distribution of the collected data [75]. To test whether the duration of seeds' soaking in the disinfection solution had an influence on the contamination, death (seed burning), and germination rate [76]—as well as the influence of substrate type on the acclimatisation of *S. longepedunculata* vitroplants—a one-way analysis of variance (ANOVA) at the 5% level was performed using R software. A

factorial two-way ANOVA was used to evaluate the effects of two independent factors—namely cytokinin type and dose—on the in vitro regeneration capacity. The test was also used to evaluate the effect of auxin type and microcutting on the rhizogenesis process of *S. longepedunculata*. The mean values that showed significant differences were subjected to the Tukey test for pairwise comparison of means.

## 3. Results

### 3.1. Sanitisation of S. longepedunculata Seeds after Soaking at Different Times in Sodium Hypochlorite (10%) and Ethanol (70%)

The results of the ANOVA showed significant differences in the seed contamination and death rate under different seed soaking times ($p = 0.023$; Table 2). Indeed, the contamination rate varied from $13.2 \pm 1.7\%$ to $36.5 \pm 19.1\%$. The seeds with a soaking time in ethanol of less than 5 min had a high contamination rate ($36.5 \pm 19.1\%$), whereas those with a soaking time in ethanol of 5 min showed some decontamination efficiency, with a contamination rate lower than 20%. Finally, a low seed death rate ($9.9 \pm 5.9\%$) was recorded when *S. longepedunculata* seeds were soaked for 15 min in NaOCl (10%) and then in $C_2H_6O$ followed by 10 min in NaOCl (10%). Conversely, the highest seed death rate ($33.2 \pm 22.3\%$) was obtained with a 10 min soak in 10% NaOCl followed by a 10 min soak in 70% ethanol ($p = 0.022$; Table 2). The next-highest seed death rate (of about 33%) was from a 5 min soak in 10% NaOCl followed by 10 min in 70% ethanol. This suggests that with the same soaking time (10 min) in 70% ethanol, soaking *S. longepedunculata* seeds beyond 5 min in 10% sodium hypochlorite solution slightly increased the death rate (Table 2).

**Table 2.** Contamination rate (%), death rate (%), and germination rate (%) of *S. longepedunculata* seeds after disinfection in different solutions. Each treatment consisted of twenty-four replicates. Means followed by the same letter within each column are not significantly different according to Tukey's test at $p < 0.05$.

| Treatment | Contamination Rate (%) | Dead Rate (%) | Germination Rate (%) |
|---|---|---|---|
| Treatment 1 | $33.2 \pm 22.3$ a | $19.8 \pm 17$ ab | $50 \pm 17.9$ ab |
| Treatment 2 | $36.5 \pm 19.1$ a | $13.2 \pm 17$ ab | $50.1 \pm 21.2$ ab |
| Treatment 3 | $29.9 \pm 24.6$ a | $26.6 \pm 26.3$ ab | $46.7 \pm 23.5$ b |
| Treatment 4 | $13.2 \pm 1.7$ b | $9.9 \pm 5.9$ b | $76.8 \pm 22.4$ a |
| Treatment 5 | $13.2 \pm 1.7$ b | $36.5 \pm 19$ a | $50.0 \pm 17.9$ ab |
| Treatment 6 | $13.3 \pm 2.3$ b | $33.2 \pm 22.3$ ab | $53.5 \pm 23.5$ ab |
| $p$ | 0.023 | 0.022 | 0.029 |

Treatment 1: 20 min in 10% sodium hypochlorite and 30 seconds in 70% ethanol; Treatment 2: 15 min in 10% sodium hypochlorite and 1 min in 70% ethanol; Treatment 3: 15 min in 10% sodium hypochlorite and 2 min in 70% ethanol; Treatment 4: 10 min in sodium hypochlorite 10% and 5 min in ethanol 70%; Treatment 5: 10 min in sodium hypochlorite 10% and 10 min in ethanol 70%; and Treatment 6: 5 min in sodium hypochlorite 10% and 10 min in ethanol 70%.

A discrimination between mean values was obtained from an ANOVA on the germination rate of *S. longepedunculata* seeds under the influence of disinfection time ($p = 0.029$). None of the treatments are significantly different except Treatment 3, which is significantly worse than Treatment 4. Indeed, seeds disinfected with NaOCl (10%) coupled with $C_2H_5OH$ (70%) for 10 and 5 min of soaking, respectively, had a better germination rate ($76.8 \pm 22.4\%$), whereas soaking for 15 min in NaOCl (10%) and then 2 min in $C_2H_5OH$ (70%) had a $46.7 \pm 23.5\%$ germination rate—the lowest value (Table 2).

### 3.2. Proliferation of S. longepedunculata Plantlets after Enrichment of the Culture Medium with Different Types and Doses of Cytokinins

The two-factor ANOVA results for the interaction between the type and concentration of cytokinins showed significant differences in the number of plantlets (Table 3), but not in the germination rates. Overall, the obtained results show that the number of plantlets was high with a concentration of cytokinin of 1.5 mg/L (11.69, 12.04, and 12.21, respectively, with BAP, Kin, and mT) and 2 mg/L (10.82, 11.47, and 11.91, respectively, with BAP, Kin,

and mT). Regardless of the type of cytokinin, a low number of plantlets (6.87 ± 0.8 plantlets for kinetin, 7.45 ± 1.4 plantlets for 6-benzylaminopurine, and 7.52 ± 1.1 plantlets for meta-topolin) was recorded at concentrations of 0.5 mg/L. In contrast, the medium culture without cytokinins yielded one plantlet (1.2 ± 0.4) per seed sown (Table 3; Figure 1).

**Table 3.** Germination rate (%) and number of plantlets produced by seeds of *S. longepedunculata* under the influence of cytokinin type and concentration. Each treatment consisted of six replicates. Means followed by the same letter within each column are not significantly different according to Tukey's test at *p* < 0.05.

| Type of Cytokinin | Concentration of Cytokinin (mg/L) | Germination Rate (%) | Plantlets Number |
|---|---|---|---|
| Without | 0 | 79.17 ± 41.4 | 1.2 ± 0.4 g |
| BAP | 0.5 | 91.67 ± 28.23 | 7.45 ± 1.4 f |
| | 1 | 95.83 ± 20.41 | 9.61 ± 1.2 e |
| | 1.5 | 95.83 ± 20.41 | 11.69 ± 1.4 abc |
| | 2 | 95.83 ± 20.41 | 10.82 ± 1.6 bcde |
| Kin | 0.5 | 95.83 ± 20.41 | 6.87 ± 0.8 f |
| | 1 | 95.83 ± 20.41 | 10.21 ± 1.2 de |
| | 1.5 | 91.67 ± 28.23 | 12.04 ± 1.3 ab |
| | 2 | 95.83 ± 20.41 | 11.47 ± 1.2 abcd |
| mT | 0.5 | 95.83 ± 20.41 | 7.52 ± 1.1 f |
| | 1 | 91.67 ± 28.23 | 10.63 ± 1.1 cde |
| | 1.5 | 95.83 ± 20.41 | 12.21 ± 1.5 a |
| | 2 | 95.83 ± 20.41 | 11.91 ± 1.04 abc |
| *p* | | 0.74 | 0.000 |

BAP: 6-benzylaminopurine; Kin: kinetin; mT: meta-topolin; 0.5: 0.5 mg/L; 1: 1 mg/L; 1.5: 1.5 mg/L; and 2: 2 mg/L.

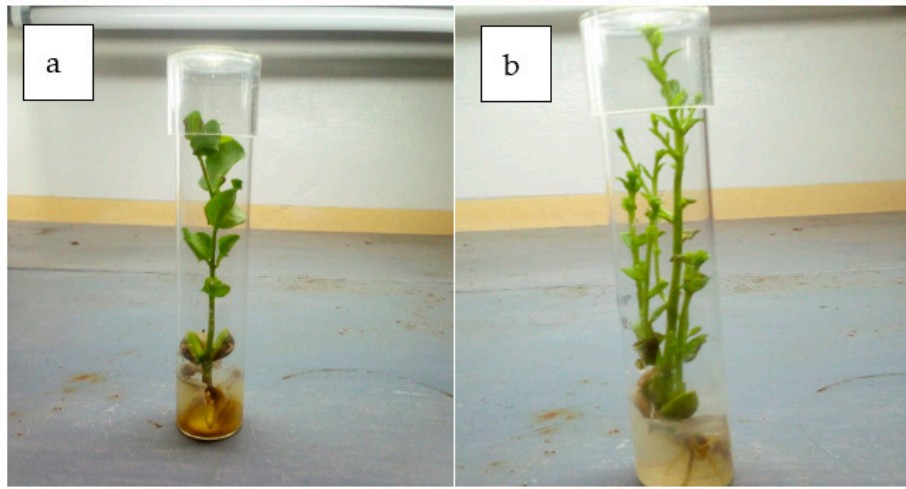

**Figure 1.** In vitro regeneration of *S. longepedunculata* in hormone-free culture medium (**a**) and in cytokinin-enriched (meta-topolin 1.5 mg/L) medium, inducing plantlet proliferation (**b**). The tubes are placed on a grey-bottomed shelf in the culture chamber.

*3.3. Recovery and Rooting of S. longepedunculata following Microcutting Culture-Medium Enrichment with Different Types and Doses of Auxins*

The results of the recovery rate, rooting rate, number of roots per microcutting, and root length due to the interaction between auxin type and microcutting part showed significant differences in the two-factor ANOVA (*p* = 0.001, *p* = 0.000, *p* = 0.000, and *p* = 0.012, respectively). For the rooting rate, the results obtained showed that all treatments were similar, with the exception of basal cuttings combined with IAA (90.00%), which was

better than basal cuttings, whatever the type of IAA (35%, 32.5%, and 30%, respectively with IAA, NAA, and IBA). Furthermore, whatever the type of auxin, all treatments showed similar performance, with the exception of apical cuttings, which showed poor performance in terms of root number (between 38.57 and 47) and length (1.33 cm, 1.66 cm, and 1.5 cm, respectively, with IAA, NAA, and IBA) compared with basal cuttings. Finally, although the ANOVA results showed significant differences between treatments, the results obtained show similar values (around 1) for rooting rate (Table 4).

**Table 4.** Recovery rate (%), rooting rate (%), length of root, and number of roots of *S. longepedunculata* microcuttings under the influence of auxin type: (IAA: indole-3-acetic acid, NAA: 1-naphthalene acetic acid, and IBA: indole-3-butyric acid) and the part of the cutting (apical, median, and basal). Each treatment consists of six replicates. Means followed by the same letter in each column are not significantly different according to the Tukey test at $p < 0.05$.

| Auxin Types | Cutting Part | Rooting Rate (%) | Number of Roots | Length of Roots (cm) | Rooting Rate (%) |
|---|---|---|---|---|---|
| IAA | Apical | 35.00 ± 27.77 b | 38.57 ± 20.35 c | 1.33 ± 0.51 c | 0.79 ± 0.40 bc |
| | Median | 50.00 ± 33.81 ab | 48.00 ± 24.66 bc | 1.83 ± 0.40 bc | 1.04 ± 0.19 ab |
| | Basal | 90.00 ± 10.69 a | 82.50 ± 15.81 a | 2.75 ± 0.46 a | 1.22 ± 0.21 a |
| NAA | Apical | 32.50 ± 23.75 b | 45.00 ± 12.25 bc | 1.66 ± 0.51 c | 0.75 ± 0.22 b |
| | Median | 42.50 ± 36.15 ab | 64.00 ± 16.73 abc | 1.60 ± 0.54 c | 0.75 ± 0.25 b |
| | Basal | 75.00 ± 33.38 ab | 85.71 ± 15.12 a | 2.57 ± 0.53 ab | 1.07 ± 0.31 ab |
| IBA | Apical | 30.00 ± 33.81 b | 47.50 ± 15.00 bc | 1.50 ± 057 c | 0.81 ± 0.23 abc |
| | Median | 42.50 ± 29.15 ab | 73.33 ± 10.33 ab | 1.83 ± 0.40 bc | 0.91 ± 0.30 abc |
| | Basal | 72.50 ± 33.70 ab | 85.71 ± 15.12 a | 2.71 ± 0.48 ab | 1.21 ± 0.36 a |
| *p* | | 0.001 | 0.000 | 0.000 | 0.012 |

### 3.4. Acclimatisation of S. longepedunculata Vitroplants

The recovery rate of seedlings ranged from 85.00 ± 16.79% (agrarian soil + undisturbed soil) to 85.63 ± 20.32% on agrarian soil, with similar values to those recorded on undisturbed soil. However, the agrarian soil yielded low-growing seedlings of 15.71 ± 2.07 cm, whereas the high growth performance of *S. longepedunculata* plants was noted on undisturbed soil at 18.95 ± 2.07 cm. Despite this, the results of the ANOVA showed no significant difference between the types of substrates on all parameters observed in the acclimatisation phase on *S. longepedunculata* vitroplants (Table 5).

**Table 5.** Recovery and growth of *S. longepedunculata* vitroplants microcut under the influence of substrate type in the nursery. Each treatment consisted of twelve replicates. The presence of identical letters in the same column means that there was no difference between the treatments according to an analysis of variance at a 5% threshold.

| Substrate Types | Recovery Rate (%) | Height of Plantlets (cm) |
|---|---|---|
| Agrarian soil | 85.63 ± 20.32 a | 15.71 ± 2.07 a |
| Undisturbed soil | 85.63 ± 20.32 a | 18.95 ± 2.07 a |
| Undisturbed soil + agrarian soil | 85.00 ± 16.79 a | 16.72 ± 2.11 a |
| *p* | 0.986 | 0.498 |

## 4. Discussion

### 4.1. What In Vitro Regeneration Protocol Is Best for S. longepedunculata, a Species Overexploited for Its Medicinal Virtues in the Lubumbashi Region?

For its best expression in in vitro culture, the explant should be free of pathogens, and several disinfection solutions are used for this purpose [70,77]. Since no single disinfectant is able to kill the pathogens that can develop on the propagating organs, several types of disinfectant in various proportions are used. We found that soaking *S. longepedunculata* seeds in ethanol for less than five minutes resulted in a contamination rate of 30%, while

longer durations of soaking in ethanol (10 min) resulted in a loss of seeds due to burning. In line with our results, Lijalem et al. [47] revealed that increasing Clorox concentration increased the decontamination but decreased the viability of *S. longepedunculata* seeds in in vitro propagation. Indeed, the extension of the soaking time beyond five minutes becomes harmful for the seeds because of the toxicity of the solution. Guanih et al. [78] found that sterilisation of seeds of the medicinal plant *Dryobalanops lanceolata* with 50% Clorox concentration for more than 20 min broke the seed coat and reduced seed viability. Badou et al. [24] and Contreras-Loera et al. [79] stated that when soaking the explant in a sanitising solution, a short soaking time led to contamination, while a long soaking time led to burning—two factors responsible for the reduction in the germination rate. The results we obtained on the contamination and death rate of *S. longepedunculata* confirm our hypothesis that an intermediate soaking time (between 5 and 10 min) would be the most suitable for the sanitation of *S. longepedunculata* seeds.

Furthermore, cytokinins play various roles during plant development, acting in collusion with other hormones—preferentially auxins—on the regulation of cell division and differentiation [80]. Thus, the addition of cytokinin quantities to the culture medium in this study allowed a significant proliferation of axillary stems in *S. longepedunculata* [19]. This result confirms the postulation of Bultynck and Lambers [81] that cytokinins induce cell division, one of the physiological functions necessary for plant propagation. Furthermore, when using cytokinins or any other hormones, two factors influence the response of the explant: namely, the form (type) and the quantity supplied [81]. Lower doses of exogenously applied cytokinins increased the activation of mitogen-associated protein kinase, while higher doses decreased its activation, suggesting a balanced level of cytokinins is required to increase plantlet production [82,83]. Our results also show that meta-topolin resulted in a slightly higher number of plantlets, corroborating the results of the micropropagation of *A. polyphylla* seedlings [84]. Cara et al. [85] obtained an increase in the number of axillary stems in *C. spinosa* by enriching the culture medium with 1.4 mg/L of a cytokinin and not at any other concentration. Mazinga et al. [86] also obtained better proliferation of banana shoots in in vitro culture using meta-topolin. Indeed, the response of plants to different types and concentrations of growth regulators varies because of differences in their endogenous levels of growth regulators [47]. Our results confirm that an intermediate concentration of cytokinins induces a significant proliferation of axillary stems in *S. longepedunculata*, and that meta-topolin promotes greater proliferation compared to other cytokinins.

No significant difference was obtained in the recovery and growth in the size of *S. longepedunculata* seedlings on different substrates. This result is similar to that obtained by Mazinga et al. [56] on *Moringa oleifera* Lam., suggesting that plant recovery is more dependent on material quality and climatic conditions, such as water [87] or temperature [88]. Indeed, the similar height growth of *S. longepedunculata* vitroplants, regardless of the type of substrate, is justified by the fact that plantlets were grown on unique soil types (Ferralsols), one of the main characteristics of which is the deficiency of major mineral elements [89].

Obtaining adventitious roots is the major constraint to be overcome in order to achieve successful plant cuttings [90]. Indeed, the appearance of roots on cuttings is a unique and complex process [91]. In *S. longepedunculata*, this process varied depending on the part of the microcutting produced: basal cuttings had a higher recovery rate than the median and apical cuttings. Henselova et al. [92] also observed better recovery of basal cuttings due to their physiological maturity and thermal plasticity, which facilitates adaptability to recovery. De Klerk [93] reported that, depending on the species, the basal part lends itself to easy rooting, which is also confirmed by our results. However, our results highlight the absence of a preference of *S. longepedunculata* for a particular auxin type, which contrasts with Ansar et al. [94], who noted a better performance of indole-3-butyric acid in *Olea europaea* L. root proliferation. Pierik [95] reported that culture medium supplemented with auxin at low concentrations in combination with cytokinin promoted the growth and formation of new shoots, consequently increasing the plantlet multiplication rate.

*4.2. Conservation Implications*

In the Lubumbashi region, there is a currently significant increase in people using medicinal plants, including *S. longepedunculata*, for the relief of various diseases [95–97]. Such heavy reliance on *miombo* woodland for the collection of therapeutic plants—coupled with other practices, such as cutting of trees for charcoal production, urbanisation, and mining activities, mortgages the existence of the forest [31,34,98]. The ability to propose a rapid propagation protocol for a forest species that has a low regeneration capacity in the wild is important for the sustainability of its populations. Indeed, as *S. longepedunculata* is highly prized in Lubumbashi, its rapid regeneration would allow it to be integrated into a program restoring degraded landscapes [9,99,100]. A large number of plantlets obtained from microcutting can serve as explants for rhizogenesis induction [10,69,101]. Furthermore, after root production, these seedlings will be acclimatised and eventually used in the reforestation program. They can also be associated with crops: this option is particularly interesting, as *S. longepedunculata* leaf litter may be used to increase crop yields [102], as well as increase farmers' incomes [103]. In addition, with the constant regression of green wooded areas in Lubumbashi [52], urban forestry is expanding into residential plots through the planting of ornamental and utility species, including those of medicinal value [104,105]. *S. longepedunculata* seedlings can therefore be planted in plots as a local source of fresh medicinal roots for the community, thus better preserving their active ingredients. Furthermore, studies have shown the potential of in vitro regeneration to improve the concentration of secondary metabolites in wild medicinal plants [106], which may broaden their pharmacological applications. In an environment where most foodstuffs sold contain high levels of toxic metals [107], in vitro micropropagation also appears to be a low-cost way to limit the soil-to-plant transfer of toxic metals [108].

Our study of the in vitro multiplication of *S. longepedunculata*—while potentially very useful and beneficial—also has certain limitations. The first relates to cost and complexity. Indeed, in vitro multiplication techniques can be costly and require specialised equipment, facilities, and skills [108,109]. Secondly, in vitro cultures are susceptible to microbial contamination, which can compromise the purity of cultures and the validity of results [110]. Thirdly, in vitro propagation can lead to a reduction in genetic diversity compared with the mother plants, which may have implications for the genetic stability and medicinal efficacy of plants [111]. Fourthly, in vitro culture conditions can be very different from the natural conditions under which plants grow [112]. This can lead to differences in the biochemical and medicinal properties of plants produced in vitro compared with those found in their natural environment [113]. Finally, the transfer of results to large-scale production is also a challenge, as cultural conditions need to be adapted to ensure consistent yield and quality [114]. Despite these limitations, in vitro propagation remains a valuable tool for the rapid and controlled propagation of plant species of medicinal interest [115]. However, it is essential to recognise these limitations and take them into account when assessing the relevance and applicability of the results of such a study [116].

## 5. Conclusions

This study aimed to achieve rapid proliferation of *S. longepedunculata* plantlets by elucidating the optimal type and effective amount of cytokinins after seed disinfection. This was conducted through two trials and under a completely randomised design.

The obtained results confirm that the average soaking time of *S. longepedunculata* seeds in a disinfection solution allows for efficient sanitation. Soaking for 10 min in NaOCl followed by 5 min in ethanol was identified as the most effective method, as it increased the germination rate by significantly reducing the rates of contamination and scorching. Furthermore, the addition of intermediate doses of cytokinins increased the proliferation of plantlets from a single explant, while meta-topolin slightly increased the number of plantlets compared to other types of cytokinin at the same dose. The culture medium enriched with 1.5 mg/L meta-topolin promoted the induction of a high number of *S. longepedunculata* plantlets. However, no significant differences were obtained with the three auxin types in

the rhizogenesis process. Microcuttings of the basal parts of the stem had a higher recovery rate, rooting rate, number of roots per microcuttings, and mean root length than the apical parts. The results of the ANOVA showed no significant difference between the types of substrates on all parameters observed in the acclimatisation phase on *S. longepedunculata* vitroplants.

By producing many plantlets from a single seed, this study opens the way for the rapid proliferation of *S. longepedunculata* with the use of plantlets as explants in rhizogenesis. The seedlings produced can be used to restore degraded landscapes, used in association with crops to increase yield and farmer income, or installed in residential plots.

**Author Contributions:** Conceptualisation, M.C.M. and M.M.K.; methodology, M.C.M., S.W., Y.U.S. and M.M.K.; software, S.W. and M.C.M.; validation, all authors; formal analysis, S.W. and M.C.M.; investigation, M.C.M.; resources, M.C.M., J.B., S.W., Y.U.S. and M.M.K.; data curation, M.C.M., J.B., S.W., Y.U.S. and M.M.K.; writing—original draft preparation, M.C.M.; writing—review and editing, all authors; visualisation, M.C.M. and Y.U.S.; supervision, S.W., Y.U.S. and M.C.M.; project administration, S.W., Y.U.S. and M.M.K.; funding acquisition, M.C.M., Y.U.S. and J.B. All authors have read and agreed to the published version of the manuscript.

**Funding:** This research was funded by the IUC project "Challenges and opportunities for a sustainable socio-ecology in the Katangese Copperbelt Area" (VLIR-UOS) and the development research project "Capacity Building for the Sustainable Management of the miombo woodlands through the Assessment of the Environmental Impact of Charcoal Production and the Improvement of Forest Resource Practices" (PRD CHARLU, ARES-CCD).

**Data Availability Statement:** Data is contained within the article.

**Conflicts of Interest:** The authors declare no conflict of interest.

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
