# Peer review of "In Vitro Regeneration Protocol for Securidaca longepedunculata Fresen., a Threatened Medicinal Plant within the Region of Lubumbashi (Democratic Republic of the Congo)"

_conservation, doi:10.3390/conservation3030028_

Round 1

Reviewer 1 Report

Although the manuscript does have some valuable scientific information, but the presentation needs improvement to make it publishable. Please consider following points for improvement. 

Keywords should not repeat words from the title. 

The introduction is too long and contains redundant information irrelevant to the study.  Many of the references are not verified and/or are in a language other than English. 

The Materials and Methods section is not a section for discussion. Also, irrelevant information should be removed. Language should be improved for clarity. 

The results data presentation in Tables 3 and 4 is unscientific, as the authors have combined the data from different treatments and have compared it with the control.     

 Discussion should be focused on the limited scope of this study. 

Overall presentation and language of the manuscript must be improved to make it publishable. 

Language must be improved for clarity.  

Author Response

Keywords should not repeat words from the title. 

Keywords have been changed as follow:

Keywords: miombo woodland; medicinal plant; plant growth regulators; culture medium; micro-cuttings; agroforestry.

The introduction is too long and contains redundant information irrelevant to the study.  Many of the references are not verified and/or are in a language other than English. 

The introduction has been shortened by removing redundant elements. References have been checked and only the most relevant have been retained.

The Materials and Methods section is not a section for discussion. Also, irrelevant information should be removed. Language should be improved for clarity. 

Certain elements of the discussion arguing the various choices made in the methodology have been removed, while remaining supported solely by the references.

The results data presentation in Tables 3 and 4 is unscientific, as the authors have combined the data from different treatments and have compared it with the control.     

Both tables have been deleted.

 Discussion should be focused on the limited scope of this study. 

A section on methodological limitations has been added at the end of the discussion (in the conservation implications section).

Overall presentation and language of the manuscript must be improved to make it publishable. 

The language has been corrected

Comments on the Quality of English Language

Language must be improved for clarity.  

Language has been corrected.

Reviewer 2 Report

Dear Authors, please see the comments regarding your manuscript below.

1. Expand on the ecosystem services provided by forests and emphasis the importance of these.

2.      If you state statistics, kindly include the actual numbers/percentage of reforestation.

3.      Elaborate on the shortfalls of reforestation compared to micro-propagation.

4.      Correct spacing on line 71.

5.      Please explain how the male population is exposed to toxic metals? Furthermore, exposure to toxic metals has several other health effects, not only erectile dysfunction. Please state and explain effects of all.

6.      Kindly include a reference/citation for each sentence. There are many sentences that lack reference to literature. Correct throughout.

7.      Please give coordinates of the study sites.

8.      During which seasons/months was the study conducted?

9.      Give exact pH of the soil conditions.

10.   Since the soil contains high FE and Al, were there any controls conducted in this study using soil with appropriate nutrients and other acceptable metals?

11.   I am concerned that the high Fe and Al might affect the overall results of this study.

12.   How were the plants analysed for pathogens?

13.   Apply italics to the species name in line 178

14.   What solution was used to water the plants, and how often was this done?

15. The results and discussion are well-written however requires updated and additional references. 

Minor editing of English required. 

Author Response

  1. Expand on the ecosystem services provided by forests and emphasis the importance of these.

Added like this: The importance of the ecosystem services provided by forests is hard to overstate. They contribute to global climate stability, support biodiversity, protect soil and water resources, provide livelihoods for local communities, offer spaces for recreation and meditation, and play an essential role in the ecological resilience of our planet. The conservation and sustainable management of forests is therefore essential to preserve these services and ensure a sustainable future for generations to come.

  1. If you state statistics, kindly include the actual numbers/percentage of reforestation.

In the Lubumbashi plain, with a total landscape area of 8877km², a 7% loss of miombo woodland was observed between 2005 and 2011, compared with a 3.7% regeneration.

  1. Elaborate on the shortfalls of reforestation compared to micro-propagation.

Added like this: But, in many regions, when resorting to reforestation, it can be difficult to find reliable sources of seed from native trees or trees adapted to local conditions. Indeed, native species are less used due to low seed capacity germination coupled to high mortality of seedlings. Reforestation can also reduce the genetic diversity of new plantations by using seed from a limited number of trees, and often requires the collection of seeds from natural habitats, which can lead to the disruption of existing ecosystems. Yet, micropropagation enables plants to be multiplied from a small sample of selected plant material. It also preserves the genetic diversity of the original plants, which is crucial to the resilience and adaptability of tree populations in the face of environmental change and threats such as disease and pests. Micropropagation enables rapid multiplication of plants in the laboratory, which is particularly useful for slow-growing or threatened species. It enables also seedlings to be produced without disturbing wild tree populations, thus contributing to biodiversity conservation.

  1. Correct spacing on line 71. Ok, it is done.
  2. Please explain how the male population is exposed to toxic metals? Furthermore, exposure to toxic metals has several other health effects, not only erectile dysfunction. Please state and explain effects of all.

This section has been removed from the text. However, male populations are exposed to trace metals through various artisanal mining activities (digging, transport, crushing, washing, sorting, etc.), through dust inhalation and the consumption of contaminated food.

  1. Kindly include a reference/citation for each sentence. There are many sentences that lack reference to literature. Correct throughout.

The reference/citation has been added for each sentence.

  1. Please give coordinates of the study sites.

Added as follow:

This study was conducted in the city of Lubumbashi city (11°27'-11°47'S et 27°19'-27°40' E), the capital of Upper Katanga Province in south-eastern DR Congo.

  1. During which seasons/months was the study conducted?

The trial which started in July and ended in October 2022, was conducted during dry season in the region.

  1. Give exact pH of the soil conditions.

The soil used during this test had a pH of 5.5

  1. Since the soil contains high FE and Al, were there any controls conducted in this study using soil with appropriate nutrients and other acceptable metals?

In our region, the dominant soils are ferralsols, with high concentrations of iron and aluminum, and poor agronomic performance. It is rare, perhaps with the exception of alluvial soils, to obtain soils that do not have these characteristics. In the absence of fertilizers, the poor performance of the seedlings was undoubtedly affected by the characteristics of acid soils: acid pH, Al and Mg toxicity, nutrient deficiencies, etc.

  1. I am concerned that the high Fe and Al might affect the overall results of this study.

In our region, the dominant soils are ferralsols, with high concentrations of iron and aluminum, and poor agronomic performance. It is rare, perhaps with the exception of alluvial soils, to obtain soils that do not have these characteristics. In the absence of fertilizers, the poor performance of the seedlings was undoubtedly affected by the characteristics of acid soils: acid pH, Al and Mg toxicity, nutrient deficiencies, etc.

  1. How were the plants analysed for pathogens?

Disinfection (applied as a preventive measure in this study) had the role of eliminating any pathogenic organism that accompanies the explant, regardless of their nature. Furthermore, the objective was not to identify the pathogens, but rather to destroy them.

  1. Apply italics to the species name in line 178

OK, corrected.

  1. What solution was used to water the plants, and how often was this done?

During the acclimatization phase, tap water was used at 20 cl per day in a 16x22 cm bag. This has been integrated into the text

  1. The results and discussion are well-written however requires updated and additional references. 

Result and discussion have been updated

Comments on the Quality of English Language

Minor editing of English required. 

English language has been corrected in the manuscript.

Reviewer 3 Report

The manuscript discussed the regeneration ability of Securidaca longipedunculata Fresen as medicinal plant. There are some major revision is needed before considering the manuscript for publication. These revisions are as follows:

The title is too long, it is better to shorten it

Results in Table 3 and Table 4 show different type of cytokines with one concentration and one type of cytokine with different concentration respectively, what is the importance of these table in the presence of table 5 in which the data show the effect of different types of the cytokines with different concentrations. Data in Table 5 is enough as data in table 3 and 4 are not obvious and not enough for clarifying the effect of different concentration of the different types of cytokines on germination

Also, Figure 1 which cytokine was used with which concentration

Similarly, the results about the rooting capacity of different explants with different hormones type data in table 8 can be enough as table 6 and 7 are not obvious and not enough for clarifying the effect of different hormones  on the root initiation from  different types of explants, beside all the data in these tables are clarified in Table 8

The work discussed the regeneration of of Securidaca longipedunculata Fresen as medicinal plant, without studying the effect of these regeneration protocols on the medicinal components of the plant. Studying the effect of these regeneration protocols on the medicinal components of the plant should be added.

Author Response

The title is too long, it is better to shorten it.

The title has been changed as follows:

In vitro regeneration protocol for Securidaca longipedunculata Fresen, a threatened medicinal plant within the region of Lubumbashi (DR Congo)

Results in Table 3 and Table 4 show different type of cytokines with one concentration and one type of cytokine with different concentration respectively, what is the importance of these table in the presence of table 5 in which the data show the effect of different types of the cytokines with different concentrations. Data in Table 5 is enough as data in table 3 and 4 are not obvious and not enough for clarifying the effect of different concentration of the different types of cytokines on germination

 Tables 3 and 4 have been deleted.

Also, Figure 1 which cytokine was used with which concentration ok this has been added to the text

Similarly, the results about the rooting capacity of different explants with different hormones type data in table 8 can be enough as table 6 and 7 are not obvious and not enough for clarifying the effect of different hormones on the root initiation from different types of explants, beside all the data in these tables are clarified in Table 8

Tables 6 and 7 have been deleted. 

The work discussed the regeneration of Securidaca longipedunculata Fresen as medicinal plant, without studying the effect of these regeneration protocols on the medicinal components of the plant. Studying the effect of these regeneration protocols on the medicinal components of the plant should be added.

Analyzes of the active principles are in progress, by comparing the plants from in vitro regeneration with those from the natural environment. Data will be submitted for publication once available.

Round 2

Reviewer 1 Report

Line 60: "It also preserves the genetic diversity of the original plants" How? This statement is not correct. 

Formatting errors must be corrected throughout the manuscript eg Line 70,    

Lines 73 and 75 Are MemTR and mTR the same compound or different? 

Why have you not italicised the species name in sub-headings? also using capital letters to start each word is also unusual. 

Table 2: the standard errors in the treatments are huge, I doubt how could this data be of any significance. None of the treatments are significantly different except treatment 3 which is significantly worse than treatment 4.

Tables 3 and 4: as per your own analysis you cannot consider any one treatment as best as statistically is not significantly different from other treatments, even though it might have a greater value. 

It is fine. 

Author Response

Line 60: "It also preserves the genetic diversity of the original plants" How? This statement is not correct. The sentence in line 60 has been deleted, and we welcome the reviewer's wise observation.
Formatting errors must be corrected throughout the manuscript eg Line 70, The formatting error has been corrected. The hook and stitch have been added after reference 24.
Lines 73 and 75 Are MemTR and mTR the same compound or different? Yes, both are part of the same compound. We've standardized the writing, keeping only MemTR.
Why have you not italicised the species name in sub-headings? also using capital letters to start each word is also unusual. The species name has been italicized in sub-heading. All words beginning with a capital letter in the middle of a sentence or subtitle have been corrected.
Table 2: the standard errors in the treatments are huge, I doubt how could this data be of any significance. None of the treatments are significantly different except treatment 3 which is significantly worse than treatment 4. The data has been rigorously collected, and the results of the statistical analyses are reported here. If the Reviewer wishes, the database and the detailed results of the statistical analyses can be shared with him. However, we have added in the interpretation of the germination results that "None of the treatments is significantly different, with the exception of treatment 3 which is significantly worse than treatment 4".
Tables 3 and 4: as per your own analysis you cannot consider any one treatment as best as statistically is not significantly different from other treatments, even though it might have a greater value.
Corrected as follows: Table 3: Overall, the obtained results show that the number of plantlets was high with the concentration of cytokinin of 1.5 mg/L (11.69, 12.04 and 12.21 respectively with BAP, Kin, and mT) and 2mg/L (10.82, 11.47 and 11.91 respectively with BAP, Kin, and mT). Regardless of the type of cytokinin, a low number of plantlets (6.87 ± 0.8 plantlets for kinetin; 7.45 ± 1.4 plantlets for 6-benzylaminopurine and 7.52 ± 1.1 plantlets for meta-topolin) was recorded at concentrations of 0.5 mg/L. In contrast, the medium culture without cytokinins yielded one plantlet (1.2 ± 0.4) per seed sown (Table 3).
Table 4: For the rooting rate, the results obtained showed that all treatments were similar, with the exception of basal cuttings combined with IAA (90.00%), which was better than basal cuttings, whatever the type of IAA (35, 32.5 and 30 respectively with IAA, NAA and IBA). Furthermore, whatever the type of auxin, all treatments showed similar performance, with the exception of apical cuttings which showed poor performance in terms of root number (between 38.57 and 47) and length (1.33cm, 1.66cm and 1.5cm respectively with IAA, NAA and IBA) compared with basal cuttings. Finally, although the ANOVA results showed significant differences between treatments, the results obtained show similar values (around 1) for rooting rate (Table 4).

Reviewer 3 Report

The manuscript is well written and discussed

Author Response

We found no comments from the reviewer.

Round 3

Reviewer 1 Report

ok 

ok